# Straight to the Gradient: Learning to Use Novel Tokens for Neural Text Generation

## Abstract

Advanced large-scale neural language models have led to significant success in many natural language generation tasks. However, the most commonly used training objective, Maximum Likelihood Estimation (MLE), has been shown to be problematic, where the trained model prefers using dull and repetitive phrases. In this work, we introduce ScaleGrad, a modification straight to the gradient of the loss function, to remedy the degeneration issues of the standard MLE objective. By directly maneuvering the gradient information, ScaleGrad makes the model learn to use novel tokens during training. Empirical results show the effectiveness of our method not only in open-ended generation, but also in directed generation. With the simplicity in architecture, our method can serve as a general training objective that is applicable to most of the neural text generation tasks.

## 1 Introduction

Text generation has been one of the most important research problems in natural language processing (NLP) (Reiter & Dale, 2000). Thanks to the advances in neural architectures, models are now capable of generating texts that are of better quality than before (Brown et al., 2020). However, despite the countless efforts that have been made to improve neural architectures, models trained with the standard Maximum Likelihood Estimation (MLE) objective are known to prefer generating dull and highly repetitive texts. For instance, in *open-ended generation* tasks, such as story continuation or open dialogue generation, it has been observed that even with large pre-trained models, *e.g.,* GPT-2 (Radford et al., 2019), high frequency tokens largely dominate the generation (Welleck et al., 2020; Holtzman et al., 2020). The same observation has been reported in *directed generation* tasks such as text summarization (Nallapati et al., 2016; See et al., 2017), image captioning (Melas-Kyriazi et al., 2018; Wang & Chan, 2019) and machine translation (Tu et al., 2016; Stahlberg & Byrne, 2019).

The methods introduced to solve the aforementioned issues with neural text generation can be primarily categorized into two groups: (*i*) *training* based methods, which include incorporating auxiliary losses (See et al., 2017; Welleck et al., 2020) and coverage vector (See et al., 2017; Tu et al., 2016); (*ii*) *decoding* based methods, such as stochastic beam search (Kool et al., 2019), top-$k$ sampling (Fan et al., 2018) and nucleus sampling (Holtzman et al., 2020).

Though decoding based methods, in particular nucleus and top-$k$ sampling, perform well in practice in open-ended generation tasks, significantly reducing degeneration problem, they do not address the fundamental issue that the token-level probabilities produced by the neural model are problematic (Welleck et al., 2020). In addition, our experiments demonstrate that sampling methods also fail to generate high-quality texts in directed generation tasks such as abstractive text summarization.

In this work, based on the known observation that the model trained with MLE objective tends to generate repititive tokens or phrases, we introduce a novel method called ScaleGrad for neural text generation training, by directly maneuvering the gradients to make the model learn to use novel tokens during training. Our method lies in the training based group, which aims to address the **fundamental modeling** problem, that is, the token-level distribution predicted by the model.

We conduct extensive experiments with different neural architectures including LSTM (Hochreiter & Schmidhuber, 1997) and Transformer (Vaswani et al., 2017) across different tasks in opened-ended and directed text generation. Through extensive analysis we demonstrate that ScaleGrad consistently improves the generation quality according to both human evaluation and automatic

metrics. Compared to other training based methods, ScaleGrad is architecturally simpler and easier to fit into current neural models (§3.2), while possessing a wider applicability to different tasks compared to decoding based methods (§4.2 and §5.2).

## 2 BACKGROUND

### 2.1 NEURAL TEXT GENERATION

The NLP tasks involving text generation can be broadly categorized into two types: *directed generation* and *open-ended generation* (Holtzman et al., 2020). In the former case, the output text can be seen as a constrained transformation of the input. Examples include text summarization, machine translation, and image captioning. In the later case, the input context only provides a certain degree of constraints such that the model is allowed to generate the following texts with a considerable degree of freedom. Story/text continuation and dialogue generation fall in this category.

Neural models frame text generation tasks as some form of conditional language modeling, which is typically trained to maximize the log likelihood (equivalently, minimize the negative log likelihood) of the training data. The *Maximum Likelihood Estimation* or MLE objective for an input-output pair $(\boldsymbol{x}, \boldsymbol{y})$ can be expressed as follows.

$$\mathcal{L}_{\text{MLE}} = -\sum_{t=1}^{T} \log p_\theta(y_t | y_{<t}, \boldsymbol{x}) \tag{1}$$

where $\theta$ denotes model parameters, $T$ is the length of the output sequence $\boldsymbol{y}$, and $\boldsymbol{x}$ is the task-specific input condition, *e.g.,* source document in summarization, image in image captioning, conversation history in dialogue generation and $\emptyset$ in text continuation. *Teacher Forcing* (Williams & Zipser, 1989), where current step's target token is passed as the next input to the decoder rather than the predicted token, is usually used to train neural text generation models for faster convergence.

**Degeneration** Degeneration has been a key problem in neural text generation models for open-ended tasks, where the model generates texts that are repetitive, overly generic (dull), incoherent and gibberish. It can happen at different levels of granularity – token, phrase, sentence and paragraph. The problem has not been mitigated even with large-scale pre-trained models like GPT-2 Large (Radford et al., 2019; Holtzman et al., 2020). Degeneration has also been observed in directed generation tasks even though the output in these tasks is confined by the input. For instance, in text summarization, most of the advanced models such as BertSum (Liu & Lapata, 2019), BART (Lewis et al., 2019) and ProphetNet (Yan et al., 2020) make use of tri-gram blocking (Paulus et al., 2018) within beam search to remove duplicate trigrams during decoding, which improves the generation quality in terms of automatic metric. This implies that even with involvement of large-scale pre-trained models, degeneration still exists. Similar issues have been reported in machine translation (Koehn & Knowles, 2017; Stahlberg & Byrne, 2019) and image-description generation (Melas-Kyriazi et al., 2018; Wang & Chan, 2019).

### 2.2 COMBATING NEURAL TEXT DEGENERATION

Out of the methods proposed to tackle neural text degeneration, top-k sampling (Fan et al., 2018) and nucleus sampling (Holtzman et al., 2020) stand out as representatives of decoding based methods and unlikelihood training (Welleck et al., 2020) as a representative training based method. During each decoding step, nucleus and top-$k$ sampling use different functions to filter the candidate tokens, thus reformalizing the probability distribution and sample the next token from the new distribution instead of maximizing the actual likelihood. Randomness brought by these sampling methods reduces duplicate tokens in the output. However, decoding strategy solely does not solve the underlying modeling problem with MLE, as pointed out by Welleck et al. (2020). Our analysis in §5.2 also reveals that sampling methods fail to generate high-quality texts in directed generation tasks.

To address the issue with MLE, neural unlikelihood (UL) training has been proposed. During training, at each decoding step $t$, UL adds an auxiliary loss to the original cross entropy loss as follows.

$$\mathcal{L}^t = \mathcal{L}_{\text{MLE}}^t + \mathcal{L}_{\text{UL}}^t = -\log p_\theta(y_t | y_{<t}) - \alpha \cdot \sum_{c \in \mathcal{C}^t} \log(1 - p_\theta(c | y_{<t})) \tag{2}$$

where $\alpha$ is a hyper-parameter and $\mathcal{C}^t$ is the set of *negative tokens* at step $t$, which is constructed by previous context tokens that are not the current token, $\mathcal{C}^t = \{y^1, \ldots, y^{t-1}\} \setminus y^t$. The auxiliary UL

loss decreases the total loss based on the "unlikely" probabilities of negative tokens, thus implicitly reducing the probability assigned to the repetitive tokens. UL training targets at improving the underlying modeling problem, which accords with our goal. Therefore, we mainly compare our method with UL training[1]. In addition, we discuss one how our method is different from UL training from the gradient perspective in §5.4.

## 3 METHODOLOGY: LEARNING TO USE NOVEL TOKENS

Training a text generation model with MLE objective treats each token in the gold (ground truth) sequence equally. With this approach, the model exhibits the tendency to generate repetitive tokens/phrases during inference. To mitigate this degeneration problem, we argue that the model should focus on *learning to use novel tokens*, rather than treating all the tokens equally.

Formally, let $\boldsymbol{y} = (y^1, \ldots, y^t, \ldots, y^T)$ be the ground-truth token sequence that the model is learning to generate in an auto-regressive manner, one token at a time. At time step $t$, we define the token $\tilde{y}_i^t$ in the vocabulary $\mathbb{V}$ as a **novel token**, if $\tilde{y}_i^t$ has not been generated before, *i.e.,* $\tilde{y}_i^t \notin \{y^1, \ldots, y^{t-1}\}$. By the definition, we have a set of novel tokens $\mathbb{S}_{\text{novel}}^t \subseteq \mathbb{V}$ at each decoding step $t$ in training, which shrinks over time as new tokens are generated (or observed) in the ground-truth sequence (see Appendix B for an illustration). Note that the shrinking set of novel tokens is equivalent to the negative tokens in UL except that it may contain the current target token $y^t$, if it was observed before. To encourage the model to focus on learning to use novel tokens, we propose an architecturally-simple method that can fit into most of the auto-regressive generation models. Our method, requiring no carefully-designed components, goes straight to the gradient analysis of the loss function.

### 3.1 GRADIENT INFORMATION IN MLE TRAINING

Let us first consider the gradient analysis of the model trained with MLE. Let $\boldsymbol{o}^t$ denote the pre-softmax scores (*i.e.,* logits) over the vocabulary at time step $t$, where $o_i^t$ is the score for the token with index $i$. Similarly, let $p_k^t = [\text{softmax}(\boldsymbol{o}^t)]_k$ represent the probability of the ground truth token with index $k$ in the vocabulary. The partial derivative of the MLE objective (Eq. 1) at time step $t$ with respect to the logit $o_i^t$ can be shown as (omitting $t$ and 'MLE' subscript for simplicity):

$$\nabla_{o_i}\mathcal{L} = \frac{\partial \mathcal{L}}{\partial p_k} \cdot \frac{\partial p_k}{\partial o_i} = p_i - \mathbb{1}(i = k) \tag{3}$$

where $p_i = [\text{softmax}(\boldsymbol{o})]_i$ (derivation is given in Appendix A). Specifically, the gradient of the loss *w.r.t.* the ground truth token logit $o_k$ is $(p_k - 1)$ and for any other token logit $o_i$ is $p_i$. As the gradient-based optimization proceeds, the gradient converges to $\epsilon$, a number that is close enough to $0$. Another interpretation is that the gradient of the loss is supposed to be close to $0$ around a (local) minimum. Therefore, to reach the minimum point, or to make the gradient close to $0$, the model would try to reduce the probability of non-ground truth token $p_i$ and increase the probability of ground truth token $p_k$ in the MLE training.

From Eq. 3, it is clear that the gradient that every token $o_i$ in the vocabulary receives is directly related to its generation probability $p_i$. Therefore, we hypothesize that directly manipulating the generation probabilities of tokens, thereby controlling their gradients, can help us achieve our goal, which is to train the model so that it is encouraged to use novel tokens.

### 3.2 OUR METHOD: SCALEGRAD

To encourage the model to learn to use novel tokens for generation, we can control the gradient to force the model to either increase the probability of novel tokens or decrease the probability for non-novel tokens. Based on this basic idea, we propose an effective training method keeping it in the simplest form. Specifically, at each decoding step of training, we re-normalize the softmax output (the probability distribution over the vocabulary) in a way such that the model is informed of the current set of novel tokens and encouraged to use them. Assuming that $\tilde{\boldsymbol{p}}^t$ is the softmax output at step $t$ and $\mathbb{S}_{\text{novel}}^t$ is the corresponding set of novel tokens at that step according to our definition, we

---

[1]Note that we focus on comparing our work with token-level UL in this work.

re-compute the probability distribution as follows (again omitting $t$ for notational simplicity):

$$p_i = \begin{cases} \dfrac{\gamma \cdot \tilde{p}_i}{\sum_{j=1}^{|\mathbb{V}|} p_j}, & \text{if } i \in \mathbb{S}_{\text{novel}} \\[3ex] \dfrac{\tilde{p}_i}{\sum_{j=1}^{|\mathbb{V}|} p_j}, & \text{otherwise} \end{cases} \tag{4}$$

where $\gamma \in (0,1)$ is the only hyper-parameter in our method that controls to what degree we want to encourage the model to focus on novel tokens; a smaller value of $\gamma$ incurs more aggressive push for using novel tokens. The effect of this change is that we directly modify the generation probability (after re-normalization) of the novel tokens with a factor of $\lambda_i$, such that $p_i = \lambda_i \cdot \tilde{p}_i$ for $i \in \mathbb{S}_{\text{novel}}$ with $\lambda_i \in (0,1)$. Similarly, we have $p_i = \alpha_i \cdot \tilde{p}_i$ for $i \notin \mathbb{S}_{\text{novel}}$ with $\alpha_i > 1$.[2] Consequently, assuming that the ground truth token is indexed with $k$, the gradient for each token has been changed to:

$$\nabla_{o_i}\mathcal{L} = p_i - \mathbb{1}(i = k) = \begin{cases} \lambda_i \cdot \tilde{p}_i - 1, & \text{if } i = k \text{ and } i \in \mathbb{S}_{\text{novel}} \\ \alpha_i \cdot \tilde{p}_i - 1, & \text{if } i = k \text{ and } i \notin \mathbb{S}_{\text{novel}} \\ \lambda_i \cdot \tilde{p}_i, & \text{if } i \neq k \text{ and } i \in \mathbb{S}_{\text{novel}} \\ \alpha_i \cdot \tilde{p}_i, & \text{if } i \neq k \text{ and } i \notin \mathbb{S}_{\text{novel}} \end{cases} \tag{5}$$

We now discuss why these changes encourage the model to use novel tokens. As mentioned, during training the model tries to decrease the gradient norm to 0 to reach a local minimum. First, for a ground truth token (*i.e.*, $i = k$), if it is also a novel token, the gradient norm $|\lambda_i \cdot \tilde{p}_i - 1|$ is pushed away from 0 so that the model has to learn to increase the probability $\tilde{p}_i$ further to reduce the gradient norm; if it is not a novel token, $|\alpha_i \cdot \tilde{p}_i - 1|$ is pushed slightly closer to 0, which still makes the model learn to predict the ground truth but with a relatively lower strength. For non-ground truth tokens (*i.e.*, $i \neq k$), when it is not a novel token, $|\alpha_i \cdot \tilde{p}_i|$ increases the gradient norm so that the model learns to assign much lower probability $\tilde{p}_i$ to reduce it. Similarly, when the token is novel but not a ground truth token, the resulting gradient norm $|\lambda_i \cdot \tilde{p}_i|$ becomes smaller, for which the model only moderately learns to decrease the probability $\tilde{p}_i$ to reduce the norm further.

While ScaleGrad is derived from the gradient analysis of neural generation models (supervised training), it shares some commonalities with policy gradient methods in Reinforcement Learning in the sense that both operate by scaling the gradient based on different needs – learning to get more reward in policy gradient and learning to generate novel tokens in ScaleGrad (Appendix C draws this connection). Also note that the notion of *novel token set* can be adapted for different purposes. For example, one can define it to be a set of *important* tokens (*e.g.,* based on TF-IDF scores) to promote *extractiveness* or *factual correctness* in summarization. We leave such explorations for future work.

## 4 EXPERIMENTS

We showcase the performance of ScaleGrad in both open-ended and directed generation tasks. To verify the effectiveness of our approach, for all the experiments below, we use exactly the same hyper-parameters (except for method-specific ones) and setup as the corresponding baseline unless stated otherwise. All the experimental details, such as model hyper-parameters, training and dataset settings regarding the reproducibility can be found in Appendix G. For qualitative assessments, we show examples of generated texts in Table 4 and more in Appendix L.

### 4.1 OPEN-ENDED GENERATION

**Setup** We consider language modeling and text auto-completion, where we compare the performance of the model trained with ScaleGrad against the models trained with MLE and unlikelihood (UL) training (Welleck et al., 2020) introduced lately to mitigate degeneration in open-ended tasks. We follow the same setup as Welleck et al. (2020). Specifically, we fine-tune the pre-trained GPT-2 (Radford et al., 2019) on Wikitext-103 (Merity et al., 2017). The maximum sequence length is fixed to 300 tokens for all the models. Each model is trained for a maximum of 35k iterations and evaluated based on the perplexity on the validation set after every 1k iterations. We report language modeling results on the testset for each model selected according to the perplexity on the validation

---

[2] Since $\alpha_i \cdot \tilde{p}_i$ and $\lambda_i \cdot \tilde{p}_i$ are new re-normalized probabilities, they are both naturally bounded in $[0,1]$.

Table 1: Results for open-ended generation tasks on the Wikitext-103 testset. **ppl**, **uniq** and **Rep/l** are computed at BPE-level and the rest are at word-level. The "↑" denotes higher value for better performance and "↓" is the opposite. Number marked with * are estimated based on the testset. The results are averaged over 3 runs with different random seeds. Full results with standard deviation are reported in Appendix F.1.

| Models | Language Modeling | | | | | Auto Completion | | | |
|---|---|---|---|---|---|---|---|---|---|
| | ppl ↓ | uniq ↑ | Rep/16 ↓ | Rep/32 ↓ | Rep/128 ↓ | Rep-1 ↓ | Rep-2 ↓ | Rep-3 ↓ | uniq-w ↑ |
| MLE | **13.241** | 12.54k | 0.234 | 0.380 | 0.619 | 0.661 | 0.500 | 0.424 | 16.83k |
| UL ($\alpha = 0.5$) | 14.390 | 12.87k | 0.223 | 0.359 | 0.586 | 0.607 | 0.429 | 0.353 | 17.98k |
| UL ($\alpha = 1.0$) | 16.062 | 13.18k | 0.212 | 0.341 | 0.558 | 0.559 | 0.363 | 0.291 | 19.11k |
| SG ($\gamma = 0.2$) | 14.203 | **13.61k** | **0.197** | **0.317** | **0.522** | **0.443** | **0.215** | **0.143** | **22.25k** |
| Human | - | 18.27k | 0.177 | 0.285 | 0.480 | 0.382* | 0.096* | 0.037* | 27.55k* |

set. The same saved models are also used for text auto-completion, where 50 BPE (Sennrich et al., 2016) tokens (from testset) are given as prefix and the models are to generate the continuation of 100 next tokens. Following Welleck et al. (2020), we apply greedy decoding in all our experiments in this section. This allows us to evaluate the modeling capability exclusively. Later, in §5.1, we analyze how our method performs with different decoding methods in open-ended generation.

In language modeling, we measure the generation quality by the standard perplexity (ppl), and Rep/$l$ and 'uniq' measures of token-level distribution as Welleck et al. (2020). Rep/$l$ measures the number of times that a token from the previous $l$ tokens is repeated, when generating the following token; in our case, $l \in \{16, 32, 128\}$. The 'uniq' is defined as the number of unique next-token predictions on a test/validation set. For auto-completion, we report the repetition ratios of n-gram words (Rep-n) as well as the number of unique words (uniq-w) that are used during generation on the testset.

**Results** From the results in Table 1, we notice that in language modeling, the model trained with ScaleGrad (SG) yields a token distribution that is much closer to human, while maintaining a lower perplexity. In particular, compared to the best baseline, SG achieves 1%, 2%, 4% lower repetitions in Rep/16, Rep/32 and Rep/128, respectively, while having 11% lower perplexity. It also uses more **uniq**ue tokens compared to others (*e.g.,* 3% more compared to UL training). Overall, our method significantly improves the token-level distribution and keeps a high generation quality. In auto-completion, from the quantitative perspective, SG produces texts with much fewer repetitive n-grams compared to MLE and UL. It uses nearly 5.5k more unique words compared to the MLE baseline.

**Human evaluation** We have conducted a user study to verify the quality of generated texts. The study is conducted for two pairs of systems (SG vs. UL, SG vs. MLE). For each pair, we randomly choose the same 100 prefixes for the systems to produce their own continuations and ask two native speakers of English to judge which text is the better continuation of the given prefix in terms of their *relevance* to the prefix, *grammaticality* and *readability*. More details can be found in Appendix D.

From the results in Table 2, we can observe that the texts produced by the models trained with ScaleGrad (SG) are preferred by the human users in most of the cases, *i.e.,* 84.0% and 70.5%, respectively. We also compute the percentage agreement and chance-correlated Gwet's AC1/gamma coefficient (Gwet, 2008) as inter-user agreement to verify the reliability of the study (details in Appendix D). We see that the agreements are substantial in both measures.

**Generalizability** We further verify the generalizability of SG by evaluating the WikiText-103 fine-tuned model on PTB (Marcus et al., 1993) and IMDB (Maas et al., 2011) LM datasets. From the results in Appendix F.2, we see that SG generalizes well to these domains. We also show that SG consistently improves degeneration with varied decoding lengths (Appendix F.3).

Table 2: Human evaluation results for auto-completion. **% Agr.** is the percent agreement and **AC1** denotes Gwet's AC1/gamma coefficient. Winners are marked in **bold**.

| | Win Rate | % Agr. | AC1 |
|---|---|---|---|
| **SG** vs MLE | 84.0% | 84.0% | 0.78 |
| **SG** vs UL | 70.5% | 79.0% | 0.64 |

Table 3: Results for image paragraph captioning.

| Models | CIDEr |
|---|---|
| MLE w/o 3-block | 10.51 |
| UL w/o 3-block ($\alpha$=0.5) | 14.65 |
| SG w/o 3-block ($\gamma$=0.5) | **19.42** |
| MLE w/ 3-block | 22.77 |
| UL w/ 3-block ($\alpha$=0.5) | 22.25 |
| SG w/ 3-block ($\gamma$=0.5) | **24.62** |

Table 4: An example of greedy generated continuations for the same prefix.

| |
|---|
| **Prefix**: You's patriotism, and Mei's reflections on the quotidian are a few examples. More broadly, Du Fu's work in transforming the lushi from mere word play into "a vehicle for serious poetic utterance" |
| **MLE**: is a major contribution to the genre. The first of the three major works of the genre, was written in the early 1920s, and was the first of the three to be published in English. It was published in the United States in the United States, and was translated into French in the 1930s. The work was the first of the three |
| **UL**: is a major influence on the modern Chinese novel. The modern Chinese novel is often characterized as being "literary in nature", but it is not limited to literary genres. The modern Chinese novel has been described as "a literary genre that has been influenced by the Western literary tradition". The modern Chinese novel has been described as "a literary genre that has been influenced by the Western literary tradition" |
| **SG**: is a powerful example of how the Chinese literary tradition can be used to explore themes of nationalism and self determination. In addition to his own poetry, Du Fu wrote several works of prose. The first was a collection of short stories called "The Story of the Three Sisters", which was published in 1885. It was followed by a series of novels, including the first two volumes of the Song dynasty epic poem "The Three Sisters" |

## 4.2 DIRECTED GENERATION

For directed generation, we consider two tasks: image paragraph captioning and text summarization.

### 4.2.1 IMAGE PARAGRAPH CAPTIONING

**Setup** We use the captioning model proposed by Melas-Kyriazi et al. (2018) as the baseline, which comprises a CNN encoder that is pre-trained for object detection and a 1-layer LSTM decoder. The models are trained and evaluated on the paragraph captioning dataset, Visual Genome (Krause et al., 2017). We train the model with SG and compare it to the ones trained with MLE and UL. The performance is measured by CIDEr (Vedantam et al., 2015), which computes TF-IDF weighted n-gram overlaps between the model generated captions and the reference captions. We follow Melas-Kyriazi et al. (2018) to apply greedy inference since beam search did not yield any further gain.

**Results** Table 3 shows the CIDEr scores for different training methods on Visual Genome testset with and without tri-gram blocking (Paulus et al., 2018) during inference. Without tri-gram blocking, MLE produces texts that are full of repetitive phrases (see Appendix L for examples), which leads to a low CIDEr score. When UL or SG is incorporated, the performance has been notably improved from 10.51 to 14.65 and 19.42, respectively.[3] When tri-gram blocking is applied, our method is still capable of yielding 1.85 point improvement. This is because SG further improves the token-level degeneration on top of tri-gram blocking. In contrast, the model trained with UL has a slightly worse CIDEr score compared to the MLE baseline. We analyze n-gram level degeneration further in §5.2.

### 4.2.2 ABSTRACTIVE TEXT SUMMARIZATION

**Setup** We use the abstractive summarization model BertSum (Liu & Lapata, 2019) as our baseline, which adopts a Transformer architecture to take advantage of pre-trained BERT (Devlin et al., 2019) as the encoder. At the first stage, the encoder is trained with an extractive summarization objective (binary classification for sentence selection). At the second stage, it initializes the decoder randomly and (re)trains the entire encoder-decoder model with an abstrac-

Table 5: Experimental results for text summarization on CNN/DM and NYT50 testsets.

| Models | R-1 | R-2 | R-L | WMD-1 |
|---|---|---|---|---|
| **CNN/DM** | | | | |
| BertSum w/ MLE | 41.87 | 19.42 | 38.93 | 19.89 |
| BertSum w/ UL ($\alpha = 0.5$) | 42.03 | 19.36 | 39.09 | 20.21 |
| BertSum w/ SG ($\gamma = 0.8$) | **42.19** | **19.53** | **39.25** | **20.23** |
| **NYT50** | | | | |
| BertSum w/ MLE | 48.73 | 31.00 | 45.23 | 28.73 |
| BertSum w/ UL ($\alpha = 0.5$) | 48.54 | 30.73 | 44.99 | 28.50 |
| BertSum w/ SG ($\gamma = 0.8$) | **49.29** | **31.30** | **45.78** | **29.14** |

tive (or generative) objective. For our experiments, we take the encoder that was trained at the first stage and train the entire (abstractive) model with different training methods (MLE, UL and SG) using the default training setup on two benchmark datasets: CNN/DM (Hermann et al., 2015; Nallapati et al., 2016) and NYT50 (Durrett et al., 2016). During inference, length normalization (Wu et al., 2016), tri-gram blocking and beam search (beam size = 5) are used as in (Liu & Lapata, 2019).

---

[3]Although UL was originally proposed for open-ended generation, it is applicable to directed generation. We did the same scale of hyper-parameter searching for UL. Details can be seen in Appendix E.

We evaluate performance of the models with the standard F1-based ROUGE (Lin, 2004) scores (R-1, R-2, R-L) and a model-based evaluation MoverScore (Zhao et al., 2019), which computes the Word Mover Distance (WMD) between the reference summary and generated summary based on the representations from BERT. We report 1-gram MoverScore (WMD-1), which has been proven to have higher correlation with human than other metrics (Zhao et al., 2019).

**Results** From Table 5, we notice that on CNN/DM, the model trained with SG outperforms the models trained with MLE and UL when measured by ROUGE. In WMD-1, UL yields similar performance as ours. Both SG and UL further improve over the MLE baseline. The improvements imply that token-level degeneration may still exist even when tri-gram blocking is applied. On NYT-50, UL underperforms MLE, while our method improves in all measures. We discuss the possible reason why UL underperforms from a gradient perspective in §5.4.

## 5 ANALYSIS

In this section, we perform a series of analysis to gain more insights about our method.

### 5.1 OPEN-ENDED GENERATION

**Compatibility with decoding strategies** One advantage of our method is that it is compatible with decoding-based methods. One can choose different decoding strategies based on the specific needs. Table 6 provides the results of different decoding strategies used along with our SG training for text auto-completion (results for more varia-

Table 6: Results of different decoding strategies with ScaleGrad training for auto-completion.

| Approaches | Rep-1 | Rep-2 | Rep-3 | uniq-w |
|---|---|---|---|---|
| SG+Greedy Search | 0.441 | 0.214 | 0.144 | 22.23k |
| SG+Beam Search ($b = 6$) | 0.453 | 0.250 | 0.171 | 8.32k |
| SG+Top-$p$ ($p = 0.3$) | 0.356 | 0.107 | 0.049 | 30.48k |
| SG+Top-$k$ ($k = 40$) | 0.254 | 0.039 | 0.012 | 39.50k |

tions are in Appendix H). We observe that beam search, even with larger beam size, is not effective in mitigating the degeneration issue, which accords with the observation in (Holtzman et al., 2020). As expected, stochastic decoding, top-$k$ and nucleus (top-$p$) sampling, help to further reduce repetition. This sets good examples of combining training and decoding strategies for the task in hand.

Table 7: Summarization results (F1-based ROUGE-1 and MoverScore) for stochastic decoding on NYT50 testset.

| Models | ROUGE-1 | WMD-1 |
|---|---|---|
| Top-$p$ (p=0.3) | 45.44 | 24.61 |
| Top-$p$ (p=0.9) | 42.33 | 21.67 |
| Top-$k$ (k=40) | 41.23 | 20.70 |
| Top-$k$ (k=100) | 40.86 | 20.38 |
| Baseline | 48.73 | 28.73 |

Table 8: Degeneration analysis for image paragraph captioning with/without tri-gram blocking. Numbers in bold are closest to human.

| Models | Rep-1 | Rep-2 | Rep-3 |
|---|---|---|---|
| MLE | 0.723 | 0.587 | 0.530 |
| SG | 0.500 | 0.270 | 0.195 |
| MLE w/ 3-block | 0.575 | 0.271 | 0.094 |
| SG w/ 3-block | **0.440** | **0.146** | **0.037** |
| Human | 0.421 | 0.123 | 0.042 |

### 5.2 DIRECTED GENERATION

**Comparison with stochastic decoding** Although top-$p$ and top-$k$ sampling have been proven successful in open-ended generation, to our knowledge, none has tested them in directed generation tasks. In order to see if they could lead to the same improvements as ScaleGrad, we conduct additional experiments with the BertSum summmarization model, whose underlying language model is more mature due to the involvement of BERT, compared to the image paragraph captioning model. For the interested readers, we also provide the results of stochastic decoding on image paragraph captioning in Appendix I.

Table 7 shows the performance of BertSum trained with MLE on NYT50 testset when stochastic decoding is applied during inference. Since ROUGE-1 measures the exact 1-gram overlaps between reference and generated summaries, it may not be sufficient to evaluate the performance of stochastic decoding methods, which may generate more diverse output while conveying the same meaning. Therefore, we also report the MoverScore that is capable of considering the semantic similarity

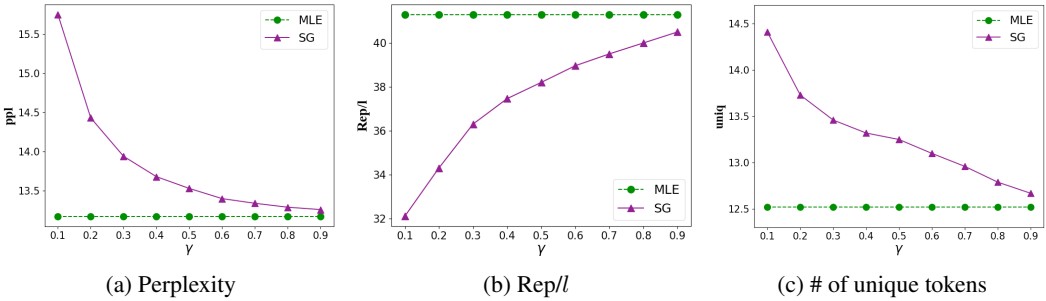

Figure 1: Hyper-parameter ($\gamma$) sensitivity in the language modeling task on Wikitext-103 development set. Rep/$l$ is computed as the average of Rep/16, Rep/32 and Rep/128. Detailed results for Rep/$l$ can be found in Appendix K.

rather than just n-gram overlaps. However, both the ROUGE and MoverScore in Table 7 lead to the conclusion that stochastic decoding methods significantly lower the performance compared to the standard beam search. This implies that they may not be a good fit for directed generation tasks. In contrast, our method possesses a wider applicability in mitigating degeneration issues.

**n-gram degeneration** To investigate further how SG minimizes degeneration and helps to improve the performance in automatic measures, we compute the n-gram repetition ratios of the outputs from the image captioning model (Melas-Kyriazi et al., 2018) and report the numbers in Table 8. [4] Compared to human, the MLE baseline has significantly higher repetitions, thus having the lowest CIDEr score (Table 3). With SG, the model yields a much better repetition ratio, which explains the notable performance boost in CIDEr. Tri-gram blocking resolves the issue of 3- or higher n-gram degeneration in a hard-coded way, improving CIDEr significantly. However, the token and 2-gram repetitions still remain high and improvable in MLE with tri-gram blocking. When both tri-gram blocking and SG are applied, the generated texts have the lowest and most human-like repetitions.

### 5.3 HYPER-PARAMETER SENSITIVITY

Towards better usage and understanding of ScaleGrad, we show how the key metrics in language modeling change with the hyper-parameter $\gamma$ in Figure 1. As discussed, a smaller value of $\gamma$ incurs a stronger push to use novel tokens, giving higher perplexity and more unique tokens. In general, $\gamma$ can be chosen based on the performance of the baseline model. If the baseline produces many repetitive tokens/phrases (*e.g.,* image paragraph captioning experiments), a smaller value of $\gamma$ should be used. Conversely, in tasks with less degeneration (*e.g.,* summarization experiments), a larger $\gamma$ can be used to further improve the unigram and bigram level degeneration without affecting the perplexity much.

### 5.4 DISCUSSION ON THE UNLIKELIHOOD TRAINING FROM A GRADIENT PERSPECTIVE

Experimental results in the directed generation tasks empirically reveal that unlikelihood (UL) training could not bring about improvements consistently. In this section, we analyze UL from the perspective of its gradients and contrast this with ours. For UL, the gradient of the total loss (Eq. 2) with a single negative token *w.r.t.* the logit $o_i$ is:

$$\nabla_{o_i}\mathcal{L} = m_i \cdot p_i - \mathbb{1}(i = k), \quad \text{where} \quad m_i = \begin{cases} (1 - \alpha\dfrac{p_{\text{neg}}}{1 - p_{\text{neg}}}) & \text{if } i \neq i_{\text{neg}} \\ (1 + \alpha) & \text{if } i = i_{\text{neg}} \end{cases} \tag{6}$$

where $p_i = [\text{softmax}(\boldsymbol{o})]_i$, $p_{\text{neg}}$ is the probability of the negative-candidate token with index $i_{\text{neg}}$, and $\mathbb{1}(i = k)$ is the indicator function with $k$ being the index of the ground truth token (see the original paper for derivation). From our previous discussion in §3.1, we know that as the gradient-based optimization proceeds, the gradient converges to $\epsilon$, a number that is close enough to $0$. Therefore, with a preset hyper-parameter, the probability of the ground truth token $p_k$ should always increase as the gradient norm of the loss *w.r.t.* its logit (*i.e.,* $|\nabla_{o_k}\mathcal{L}|$) decreases despite the ground truth token being repetitive (negative) or not. Should this not be the case, *i.e.,* the generation probability of the ground truth token $p_k$ decreases as the gradient $|\nabla_{o_k}\mathcal{L}|$ decreases, the model is not to learn

---

[4]Since Melas-Kyriazi et al. (2018) used a soft tri-gram blocking, some of the duplicate tri-grams still remain.

to predict the ground truth tokens correctly (since the $p_k$ has decreased), which in turn hurts the generation quality.

Since the ground truth is always a non-negative token by definition (*i.e.,* $i = k \neq i_{\text{neg}}$), the gradient norm from Eq. 6 is $|\nabla_{o_k}\mathcal{L}| = |\mu_k \cdot p_k - 1|$ where $\mu_k = (1 - \alpha\frac{p_{\text{neg}}}{1-p_{\text{neg}}})$. We see that when $p_{\text{neg}} > \frac{1}{\alpha+1}$ (*e.g.,* when $\alpha = 1$ and $p_{\text{neg}} > 0.5$), $\mu_k$ becomes negative, having the gradient norm $|\nabla_{o_k}\mathcal{L}| = \left| -|\mu_k| \cdot p_k - 1 \right| = |\mu_k| \cdot p_k + 1$. In this case, $p_k$ decreases as the gradient norm decreases, which contradicts with the optimization principle we mentioned earlier. To be more specific, in order to decrease the gradient norm as the training proceeds, the model will have to reduce the value of $p_k$, which goes against the goal of learning. Thus, UL becomes less effective in such special cases (subject to the choice of the value of $\alpha$). In contrast, the gradient analysis in Eq. 5 shows that ScaleGrad does not have such properties in learning to predict ground truth tokens. In our earlier exploration, we modeled the novel tokens as an auxiliary loss, which also has the similar properties as UL (Appendix J).

## 6   CONCLUSION

We have introduced a novel training method, called ScaleGrad, directly modifying the gradient of the standard MLE objective to remedy the text degeneration issues. The improvement verified by both automatic metric and human evaluation against the baselines in extensive experiments across different tasks in open-ended and directed generation and different architectures (*i.e.,* LSTM and Transformer) demonstrate the effectiveness and generalizability of our method. Further analysis shows that ScaleGrad yields token distributions that are much closer to human-written texts compared to the baselines. Our method brings a good alternative to current training strategies.

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

## A    DERIVATIONS

**Derivation of the gradient of loss *w.r.t.* logit**   We follow the same notation as in the main paper. At time step $t$, assuming that the pre-softmax scores (*i.e.,* logits) are denoted as $\boldsymbol{o}^t$ over the vocabulary $\mathbb{V}$, where $o_i^t$ denotes the score for the token with index $i$ in the vocabulary. Similarly, we have $p_i^t = [\text{softmax}(\boldsymbol{o}^t)]_i$. Let $k$ denote the index of the ground truth token at step $t$.

The cross entropy loss at step $t$ is given as (we omit $t$ for notational simplicity):

$$\mathcal{L} = -\sum_i y_i \log p_i \tag{7}$$

where $y_i = 1$ if $i = k$, otherwise $y_i = 0$. Thus the loss function can be rewritten as:

$$\mathcal{L} = -\log p_k = -\log(\frac{e^{o_k}}{\sum_j e^{o_j}}) = \log(\sum_j e^{o_j}) - o_k \tag{8}$$

Therefore, we can derive the partial derivative of the loss *w.r.t.* the logit $o_i$ as follows.

$$
\begin{aligned}
\nabla_{o_i}\mathcal{L} &= \nabla_{o_i} \log(\sum_j e^{o_j}) - \nabla_{o_i} o_k \\
&= \frac{1}{\sum_j e^{o_j}} \cdot \nabla_{o_i}(\sum_j e^{o_j}) - \mathbb{1}(i = k) \\
&= \frac{e^{o_i}}{\sum_j e^{o_j}} - \mathbb{1}(i = k) \\
&= p_i - \mathbb{1}(i = k)
\end{aligned}
\tag{9}
$$

## B    NOVEL TOKEN SET ILLUSTRATION

Figure 2 shows an example of how the novel token set changes when the model is learning to predict the sentence "people who are interested ..". At beginning, the novel token set $\mathbb{S}_{\text{novel}}$ is equivalent to the vocabulary $\mathbb{V}$. The size of the novel token set shrinks as the decoding proceeds.

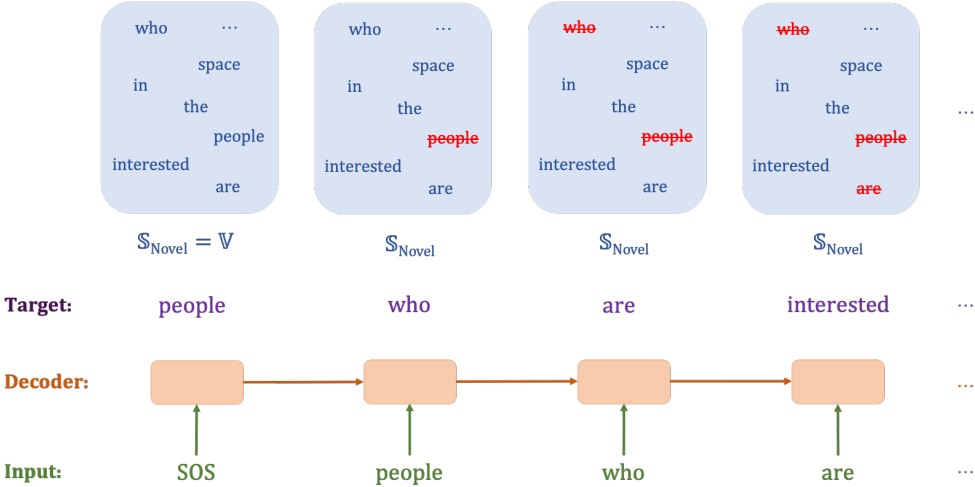

Figure 2: An illustration of how the novel token set changes as decoding proceeds for the sentence "people who are interested ...". The words marked in purple are the target words that the model is learning to predict at each decoding step.

## C  CONNECTION WITH POLICY GRADIENT OBJECTIVE IN REINFORCEMENT LEARNING

The text generation agent can also be trained with a *policy gradient* method with the objective of maximizing the expected reward (or minimizing the expected negative reward) per time-step.

$$\mathcal{L}_{\text{RL}}^t = -\mathbb{E}_{y_i^t \sim \pi_\theta} \text{r}(y_i^t) = -\sum_{y_i^t \in \mathbb{V}} p_\theta(y_i^t | y_{<t}, \boldsymbol{x}) \text{r}(y_i^t) \tag{10}$$

where $\text{r}(y_i^t)$ is the reward for token $y_i^t$ sampled from the vocabulary $\mathbb{V}$ (*i.e.,* action space) using the current policy $\pi_\theta = p_\theta(y_t | y_{<t}, \boldsymbol{x})$ and $\boldsymbol{x}$ is the input text. The policy gradient *w.r.t.* the logit $o_m$ can be expressed as follows (omitting superscript $t$).

$$\nabla_{o_m} \mathcal{L}_{\text{RL}} = -\sum_{y_i \in \mathbb{V}} \text{r}(y_i) \nabla_{o_m} \log p_\theta(y_i | y_{<t}, \boldsymbol{x}) = \sum_{y_i \in \mathbb{V}} \text{r}(y_i)(p_m - \mathbb{1}(m = i)) \tag{11}$$

Under the reinforcement learning setup, the (sampled) tokens with higher rewards will be "pushed up", or increased in probability, while tokens resulting in lower rewards will be suppressed. From the perspective of gradient analysis, Eq. 11 shows that a higher reward leads to a *larger* value of the gradient norm $|\text{r}(y_i)(p_m - \mathbb{1}(m = i))|$, which in turn forces the model to learn to assign *higher* probability $p_m$ to the the sampled token (*i.e.,* $m = i$) in order to reduce the norm $|\text{r}(y_i)(p_m - 1)|$. Meanwhile, the model also learns to assign *lower* probabilities $p_m$ to other tokens in the vocabulary (*i.e.,* $m \neq i$) to reduce the norm $|\text{r}(y_i)p_m|$.

In this specific example, reinforcement learning essentially works by scaling the gradient based on the rewards for each sampled tokens. While our method (Eq. 5) scales the gradient for each token based on the information of the novel tokens. Both of the methods share the same fundamental idea that we can have the model trained to serve the specific needs by scaling the gradient.

## D  HUMAN EVALUATION DETAILS

We conduct the human evaluation for two pairs of systems *i.e.,* SG vs. MLE and SG vs. UL. For each pair, the models generate their own continuations based on the same 100 randomly chosen prefixes. Two native speakers of English are then asked to evaluate the generated texts independently. During the study, users are instructed to judge which generated text is a better continuation of the prefix based on the overall quality (*e.g.,* readability, relevance to the prefix, grammar, and fluency).

The **Win Rate** in Table 2 is calculated as the total number of times that two users prefer the texts produced by the winner divided by the total number of cases in the evaluation ($2 \times 100 = 200$). To get a reliable human study, we also compute the percentage agreement and the chance correlated measure, Gwet's AC1/gamma coefficient (Gwet, 2008) as the inter-rater agreement. Gwet's AC1/gamma coefficient overcomes the issue where traditional measures, such as Cohen's Kappa, are not robust to skewed distributions of rankings. Figure 3 shows the interface for human evaluation study.

## E  HYPER-PARAMETER SEARCH DOMAIN FOR DIRECTED GENERATION

In the experiments with the directed generation tasks, we conduct the same scale of hyper-parameter search for unlikelihood training (UL) as our proposed ScaleGrad (SG) on the validation set. Specifically, for the hyper-parameter in length normalization (beam search decoding), we use $\beta \in \{0.0, 0.5, 1.0, 1.5, 2.0\}$ for text summarization and $\beta \in \{0.0, 0.1, 0.2, 0.3, 0.4, 0.5, 0.6, 0.7, 0.8, 0.9, 1.0\}$ for image paragraph captioning. For the model-specific hyper-parameters, $\alpha$ in UL is chosen from $\{0.5, 1.0\}$[5] and $\gamma$ in SG is chosen from $\{0.5, 0.8\}$.

---

[5]In open-ended generation, $\alpha = 1$ is recommended by the author. While in our initial exploration for directed generation, we tried other values then found that these two reduce degeneration in reasonable diverse degrees.

Figure 3: Human evaluation interface

# F    EXPERIMENTAL RESULTS ON OPEN-ENDED GENERATION

## F.1    FULL EXPERIMENTAL RESULTS ON WIKITEXT-103

We present the full experimental results on WikiText-103 (Merity et al., 2017) for open-ended generations in Table 9. All the numbers are averaged over 3 runs with different randoms seeds and shown together with standard deviations.

Table 9: Results for open-ended generations. **ppl**, **uniq** and **Rep/l** are computed at BPE-level and the rest are at word-level. The "↑" denotes higher value for better performance and "↓" is the opposite. Number marked with * are estimated based on the testset.

| Models | Language Modeling | | | | | Auto Completion | | | |
|---|---|---|---|---|---|---|---|---|---|
| | ppl ↓ | uniq ↑ | Rep/16 ↓ | Rep/32 ↓ | Rep/128 ↓ | Rep-1 ↓ | Rep-2 ↓ | Rep-3 ↓ | uniq-w ↑ |
| MLE | $13.24_{\pm 2e-4}$ | $12.54k_{\pm 4e-3}$ | $0.234_{\pm 5e-6}$ | $0.380_{\pm 8e-6}$ | $0.619_{\pm 7e-6}$ | $0.661_{\pm 1e-5}$ | $0.500_{\pm 3e-5}$ | $0.424_{\pm 7e-5}$ | $16.83k_{\pm 1e-1}$ |
| UL ($\alpha=0.5$) | $14.39_{\pm 2e-2}$ | $12.87k_{\pm 6e-3}$ | $0.223_{\pm 2e-6}$ | $0.359_{\pm 3e-7}$ | $0.586_{\pm 1e-5}$ | $0.607_{\pm 8e-5}$ | $0.429_{\pm 8e-5}$ | $0.353_{\pm 6e-5}$ | $17.98k_{\pm 4e-2}$ |
| UL ($\alpha=1.0$) | $16.06_{\pm 2e-2}$ | $13.18k_{\pm 6e-3}$ | $0.212_{\pm 1e-6}$ | $0.341_{\pm 1e-7}$ | $0.558_{\pm 9e-6}$ | $0.559_{\pm 6e-5}$ | $0.363_{\pm 2e-4}$ | $0.291_{\pm 3e-4}$ | $19.11k_{\pm 7e-2}$ |
| SG ($\gamma=0.2$) | $14.20_{\pm 2e-2}$ | $\mathbf{13.61k}_{\pm 2e-3}$ | $\mathbf{0.197}_{\pm 6e-7}$ | $\mathbf{0.317}_{\pm 1e-6}$ | $\mathbf{0.522}_{\pm 4e-6}$ | $\mathbf{0.443}_{\pm 9e-7}$ | $\mathbf{0.215}_{\pm 2e-6}$ | $\mathbf{0.143}_{\pm 4e-6}$ | $\mathbf{22.25k}_{\pm 2e-2}$ |
| Human | - | 18.27k | 0.177 | 0.285 | 0.480 | 0.382* | 0.096* | 0.037* | 27.55k* |

## F.2    ON GENERALIZABILITY OF SCALEGRAD

To further verify the generalizability (*i.e.,* different datasets and domains) of our method in open-ended generation, apart from WikiText-103 (Merity et al., 2017), we evaluate the models on two other language modeling datasets: Penn TreeBank or PTB (Marcus et al., 1993) and IMBD (Maas et al., 2011). In particular, after fine-tuning GPT-2 with different training strategies (MLE, SG and Ul) on WikiText-103 training data, we test the language modeling and auto-completion performance with the same setting described in §4.1. For PTB, we use the standard testset, while for IMDB, we randomly sample 500 movie reviews from the dataset.

Table 10 shows the experimental results on the PTB testset, from which we can see that SG consistently improves over the MLE baseline in degeneration while possessing an acceptable increase in perplexity, and it outperforms UL consistently.

Table 10: Results for open-ended generations on **PTB** testset. **ppl**, **uniq** and **Rep/l** are computed at BPE-level and the rest are at word-level. The "↑" denotes higher value for better performance and "↓" is the opposite. Number marked with * are estimated based on the PTB testset.

| Models | Language Modeling | | | | | Auto Completion | | | |
|---|---|---|---|---|---|---|---|---|---|
| | ppl ↓ | uniq ↑ | Rep/16 ↓ | Rep/32 ↓ | Rep/128 ↓ | Rep-1 ↓ | Rep-2 ↓ | Rep-3 ↓ | uniq-w ↑ |
| MLE | **33.952** | 5.60k | 0.157 | 0.292 | 0.530 | 0.652 | 0.493 | 0.424 | 6.46k |
| UL ($\alpha = 1.0$) | 41.232 | 5.96k | 0.139 | 0.260 | 0.476 | 0.533 | 0.333 | 0.259 | 7.60k |
| SG ($\gamma = 0.2$) | 40.731 | **6.15k** | **0.126** | **0.231** | **0.426** | **0.417** | **0.198** | **0.131** | **8.42k** |
| Human | - | 8.84k | 0.118 | 0.222 | 0.421 | 0.362* | 0.089* | 0.033* | 11.32k* |

Table 11: Results for open-ended generations on movie reviews from **IMDB** dataset. **ppl**, **uniq** and **Rep/l** are computed at BPE-level and the rest are at word-level. Number marked with * are estimated based on the extracted movie reviews from IMDB.

| Models | Language Modeling | | | | | Auto Completion | | | |
|---|---|---|---|---|---|---|---|---|---|
| | ppl | uniq | Rep/16 | Rep/32 | Rep/128 | Rep-1 | Rep-2 | Rep-3 | uniq-w |
| MLE | 100.764 | 7.48k | 0.153 | 0.254 | 0.449 | 0.662 | 0.499 | 0.429 | 7.70k |
| UL ($\alpha = 1.0$) | 108.334 | 8.09k | 0.123 | 0.205 | 0.373 | 0.545 | 0.346 | 0.274 | 9.31k |
| SG ($\gamma = 0.2$) | 110.451 | 8.14k | 0.114 | 0.187 | 0.344 | 0.383 | 0.142 | 0.081 | 10.42k |
| Human | - | 14.49k | 0.118 | 0.208 | 0.378 | 0.329* | 0.084* | 0.009* | *19.11k |

In Table 11, we show the experimental results on IMDB movie reviews and observe similar performance trending as in the experiment on PTB testset. From the two experiments, we can draw the conclusion that our method, SG, is capable of generalizing well to different datasets and domains. Examples of generated text for auto completion task can be found in Appendix L.

### F.3 Auto completion with different decoding lengths

In Figure 4, we show the **Rep-1** of generated text from the auto completion task with the constraints in different decoding (continuation) lengths. We observe that compared to MLE counterpart, SG yields consistent improvements on **Rep-1**, or token-level degeneration, regardless the different decoding lengths, which again verifies the effectiveness and generalizability of our method.

## G Experimental details

In this section, we present the details of the datasets used in our experiments as well as the necessary experimental setup. All the experiments were conducted with a single GPU on our machine (CPU: Intel(R) Xeon(R) Gold 6240 CPU @ 2.60GHz; GPU: NVIDIA RTX 2080Ti).

For each task in our experiments, we use the same model architecture and train it with different objectives (*i.e.,* MLE, ScaleGrad and unlikelihood). The hyper-parameters that are used for different training objectives in the same task are exactly same, except for the ones described in Appendix E. We list the key hyper-parameters in this section. Though they may not be exhaustive, all the hyper-parameters are clearly presented in our source code. In addition, all the hyper-parameters that are not listed in this section remain unchanged from their corresponding default setup.

### G.1 Open-ended generation

**Dataset** The WikiText-103 (Merity et al., 2017) is a collection of over 100 million tokens extracted from the set of verified Good and Featured articles on Wikipedia. The training, validation and test sets contain 104m, 218k and 245k tokens, respectively.

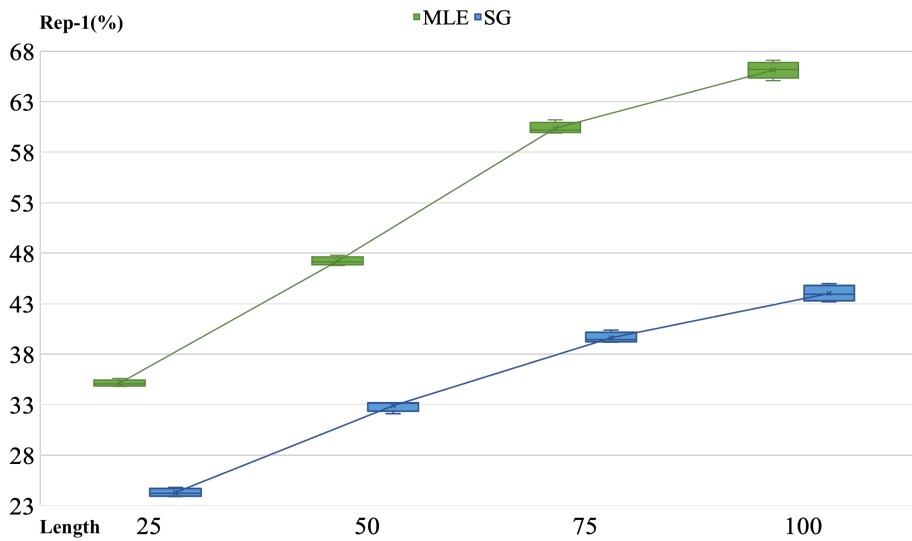

Figure 4: Box plot for **Rep-1** in auto completion with different decoding lengths. All the numbers are computed based on the results from 3 runs with different random seeds.

**Experiments**   For all the experiments, we use the same setup and the same hyper-parameters as listed in Table 12, except for the method-specific hyper-parameters. We load the GPT-2 medium and fine-tune it on WikiText-103 with a maximum of 35k iterations and select the model based on the validation perplexity.

Table 12: Hyper-parameters for open-ended generation. **M** denotes the model-specific hyper-parameters. $\mathbf{lr}_0$ is initial learning rate.

| Models | $\mathbf{lr}_0$ | **M** | **batch** |
|---|---|---|---|
| MLE | $2 \times 10^{-5}$ | – | 300 |
| UL | $2 \times 10^{-5}$ | 0.5/1.0 | 300 |
| ScaleGrad | $2 \times 10^{-5}$ | 0.2 | 300 |

### G.2   SUMMARIZATION

**Dataset**   We use CNN/DM (Hermann et al., 2015; Nallapati et al., 2016) and NYT50 (Durrett et al., 2016) in our experiments for text summarization. Table 13 shows the dataset statistics in details.

Table 13: Dataset statistics for summarization.

| **Dataset** | **Training Size** | **Validation Size** | **Test Size** |
|---|---|---|---|
| **CNN/DM** | 287,227 | 13,368 | 11,490 |
| **NYT50** | 96,834 | 4,000 | 3,452 |

**Experiments**   The models are taken from (Liu & Lapata, 2019) and we train the models for the abstractive summarization with MLE, unlikelihood training and ScaleGrad on CNN/DM and NYT50. We list the hyper-parameters that we used in Table 14.

Table 14: Hyper-parameter lists for text summarization. **M** denotes the model-specific hyper-parameters. $\mathbf{lr}_0^{\mathbf{BERT}}$ and $\mathbf{lr}_0^{\mathbf{dec}}$ stand for initial learning rate for BERT and Transformer decoder. $\beta$ is the hyper-parameter in length normalization.

| Models | $\mathbf{lr}_0^{\mathbf{BERT}}$ | $\mathbf{lr}_0^{\mathbf{dec}}$ | **M** | **batch** | $\beta$ | **Beam Size** |
|---|---|---|---|---|---|---|
| **CNN/DM** | | | | | | |
| MLE | 0.002 | 0.2 | – | 140 | 1.0 | 5 |
| UL | 0.002 | 0.2 | 0.5 | 140 | 2.0 | 5 |
| ScaleGrad | 0.002 | 0.2 | 0.8 | 140 | 1.5 | 5 |
| **NYT50** | | | | | | |
| MLE | 0.002 | 0.2 | – | 140 | 1.5 | 5 |
| UL | 0.002 | 0.2 | 0.5 | 140 | 2.0 | 5 |
| ScaleGrad | 0.002 | 0.2 | 0.8 | 140 | 1.5 | 5 |

### G.3 IMAGE PARAGRAPH GENERATION

**Dataset**  We use the image paragraph captioning corpus Visual Genome dataset, introduced by Krause et al. (2017). The dataset contains 14,575 training, 2,487 validation, and 2,489 testing images. The average length of description paragraph is 67.50 tokens.

**Experiments**  We follow the same experimental setup as in (Melas-Kyriazi et al., 2018). We train the model with different objectives and choose the model for testing based on the validation loss. During generation, tri-gram blocking and length-normalization are applied. Hyper-parameters that are used in our experiments are listed in Table 15.

Table 15: Hyper-parameter lists for image paragraph captioning. **M** denotes the model-specific hyper-parameters. $\mathbf{lr}_0$ is initial learning rate.

| Models | $\mathbf{lr}_0$ | **M** | **batch** | $\beta$ (w/o & w/ 3-blocking) |
|---|---|---|---|---|
| MLE | $5 \times 10^{-4}$ | – | 10 | 0.0/0.2 |
| UL | $5 \times 10^{-4}$ | 0.5 | 10 | 0.0/0.3 |
| ScaleGrad | $5 \times 10^{-4}$ | 0.5 | 10 | 0.6/0.6 |

## H  EXPERIMENTAL RESULTS OF DIFFERENT DECODING STRATEGIES FOR AUTO-COMPLETION.

Table 16: Results of different decoding strategies for auto-completion.

| Approaches | Rep-1 | Rep-2 | Rep-3 | uniq-w |
|---|---|---|---|---|
| Greed Search | 0.441 | 0.214 | 0.144 | 22.23k |
| Beam Search (b = 3) | 0.422 | 0.210 | 0.134 | 8.75k |
| Beam Search (b = 6) | 0.453 | 0.250 | 0.171 | 8.32k |
| Beam Search (b = 10) | 0.489 | 0.298 | 0.214 | 8.00k |
| Top-$p$ (p = 0.3) | 0.356 | 0.107 | 0.049 | 30.48k |
| Top-$p$ (p = 0.9) | 0.217 | 0.027 | 0.008 | 52.76k |
| Top-$k$ (k = 40) | 0.254 | 0.039 | 0.012 | 39.50k |
| Top-$k$ (k = 100) | 0.234 | 0.031 | 0.010 | 44.27k |

Table 16 shows the results for the auto-completion task when we train the model with ScaleGrad and infer with different decoding strategies.

# I  STOCHASTIC DECODING FOR IMAGE PARAGRAPH CAPTIONING

We apply different stochastic decoding strategies for the MLE baseline on image paragraph captioning and report the results in Table 17. The experimental results demonstrate that stochastic decoding strategies do not work well in directed generation tasks, which is consitent with our findings in summarizaiton experiments.

Table 17: Image paragraph captioning results for stochastic decoding on Visual Genome testset.

| Models | CIDEr |
|---|---|
| Top-$p$ (p=0.3) | 19.54 |
| Top-$p$ (p=0.9) | 18.67 |
| Top-$k$ (k=40) | 18.73 |
| Top-$k$ (k=100) | 18.05 |
| MLE w/ 3-block | 22.77 |

# J  NEURAL NOVEL LIKELIHOOD TRAINING

In our earlier exploration, we experimented with a regularization loss based on the novel tokens, which is similar to UL. We can call it novel likelihood (NL). The total loss at time step $t$ can be expressed as follows.

$$\mathcal{L}^t = \mathcal{L}^t_{\text{MLE}} + \mathcal{L}^t_{\text{NL}} = - \log p_\theta(y_t|y_{<t}) - \alpha \cdot \sum_{c \in \mathcal{C}^t} \log p_\theta(c|y_{<t}) \tag{12}$$

where $\alpha$ is a hyper-parameter and $\mathcal{C}^t$ is the set of *novel tokens* at time step $t$, which is the same as in ScaleGrad (§3.2), *i.e.,* $\mathcal{C}^t = \mathbb{V} \setminus \{y^1, \ldots, y^{t-1}\}$ with $\mathbb{V}$ being the vocabulary. The NL loss $\mathcal{L}_{\text{NL}}$ boosts the probabilities of novel tokens. In earlier empirical evaluation on language model, it yielded similar performance as UL. We thus also analyze the method from a gradient perspective. According to Eq. 12, it is easy to show that at time step $t$, the gradient of the overall loss *w.r.t.* the logit $o_i$ for one single novel token is (omitting $t$ for simplicity):

$$\nabla_{o_i}\mathcal{L} = p_i - \mathbb{1}(i = k) + \alpha \cdot (p_i - \mathbb{1}(i = i_n)) = (1 + \alpha) \cdot p_i - \mathbb{1}(i = k) - \alpha \cdot \mathbb{1}(i = i_n) \tag{13}$$

where $p_i = [\text{softmax}(\boldsymbol{o})]_i$, $i_n$ is the index of the novel token and $k$ is the index of the target token. We can see that when the target token is not a novel token, *i.e.,* $i = k \neq i_n$, the gradient norm becomes $|\nabla_{o_i}\mathcal{L}| = |(1 + \alpha) \cdot p_i - 1|$. When $p_i > \frac{1}{1+\alpha}$, the norm decreases as $p_i$ increases, which contradicts the gradient optimization principle. Thus, the NL loss has the similar property as the UL loss.

# K  HYPER-PARAMETER SENSITIVITY

To fully present the sensitivity of Rep/$l$ to the hyper-parameter, we further show how the Rep/$l$ (*i.e.,* $l$=16, 32 and 128) change with $\gamma$ in Figure 5.

# L  EXAMPLES

In the following, we show the examples of generated texts in three tasks: auto-completion (Table 18 and Table 19), image paragraph captioning (Table 20 and Table 21) and text summarization (Table 22, Table 23, Table 24 and Table 25). In addition, Table 26 and Table 27 show the example of auto completion on PTB testset and movie reviews from IMDB dataset.

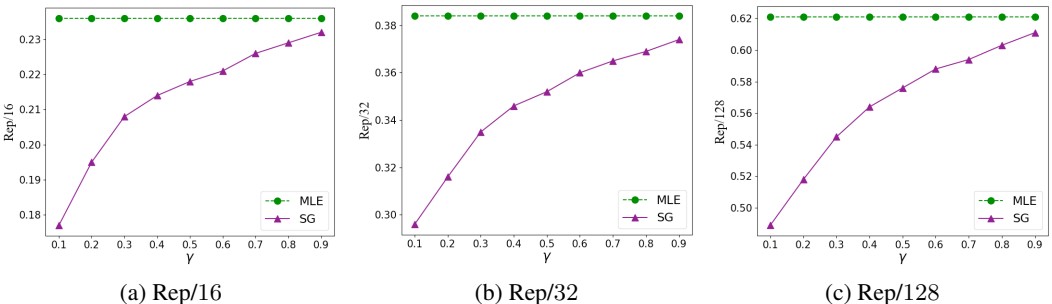

Figure 5: Hyper-parameter ($\gamma$) sensitivity in the language modeling task on Wikitext-103 development set.

Table 18: Example 1 of generated continuations for different training objectives.

| **Prefix** |
| --- |
| Robert Boulter is an English film, television and theatre actor. He had a guest starring role on the television series The Bill in 2000. This was followed by a starring |
| **MLE** |
| role in the film The Last of the Mohicans in 2002. He has appeared in the television series The X Factor, The X Factor Australia, The X Factor New Zealand, The X Factor South Africa, The X Factor New Zealand, The X Factor South Africa, The X Factor New Zealand, The X Factor New Zealand, The X Factor New Zealand, The X Factor New Zealand, The X Factor New Zealand, The X Factor New Zealand, The X Factor New Zealand, The X Factor New Zealand |
| **Unlikelihood** |
| role in the film The Last of the Mohicans, which was released in 2002. He also appeared in the television series The X Factor, where he played the role of a DJ. In 2005, he starred in the film The Last of the Mohicans, which was released in 2006. He also appeared in the television series The X Factor, where he played the role of a DJ. In 2008, he starred in the television series The X Factor, where he played the role |
| **ScaleGrad** |
| role in the BBC's comedy series The Office in 2002. In 2003, he appeared in the BBC's crime drama series The Yard, playing a police officer who is caught up in a murder investigation. He also appeared in the BBC's crime drama series The X Factor, playing a man who is accused of murdering his wife. In 2004, he starred in the BBC's comedy series The Office, playing a fictional version of himself. He has appeared in several |

Table 19: Example 2 of generated continuations for different training objectives.

| **Prefix** |
| --- |
| You's patriotism, and Mei's reflections on the quotidian are a few examples. More broadly, Du Fu's work in transforming the lushi from mere word play into "a vehicle for serious poetic utterance" |

| **MLE** |
| --- |
| is a major contribution to the genre. The first of the three major works of the genre, was written in the early 1920s, and was the first of the three to be published in English. It was published in the United States in the United States, and was translated into French in the 1930s. The work was the first of the three |

| **Unlikelihood** |
| --- |
| is a major influence on the modern Chinese novel. The modern Chinese novel is often characterized as being "literary in nature", but it is not limited to literary genres. The modern Chinese novel has been described as "a literary genre that has been influenced by the Western literary tradition". The modern Chinese novel has been described as "a literary genre that has been influenced by the Western literary tradition" |

| **ScaleGrad** |
| --- |
| is a powerful example of how the Chinese literary tradition can be used to explore themes of nationalism and self determination. In addition to his own poetry, Du Fu wrote several works of prose. The first was a collection of short stories called "The Story of the Three Sisters", which was published in 1885. It was followed by a series of novels, including the first two volumes of the Song dynasty epic poem "The Three Sisters" |

Table 20: Example 1 of generated paragraphs for different training objective on Visual Genome testset.

| **Reference** |
| --- |
| Three adult horses are in a field and a small horse. A horse color light brown has the neck low and is eating grass. This horse has the mane and the tail color tan. Two dark brown horses also are eating grass from the ground. The baby horse has a white spot on the front and is tiny. On back there is a wall of stones. |

| **MLE w/o tri-gram blocking** |
| --- |
| There are two horses in a field. The horses are brown and white. The horses are brown and white. The horses are brown and white. The horses are brown and white. The horses are brown and white. The horses are brown and white. The horses are brown. The horses are brown. The horses are brown. The horses are brown. The horses are brown. The horses are brown. The horses are brown. The horses are brown. The horses are brown . The horses are brown. The horses are brown. The horses are brown. The horses are brown. The horses are brown. The horses are brown. |

| **MLE w/ tri-gram blocking** |
| --- |
| There are two horses in a field. The horses are brown and white. The horse on the right is brown and brown. The horse is brown. The other horse is white. The grass is green and green. The sky is very and green. There are trees in the background. The trees are green and healthy. The horses are black. The sun is shining brightly. |

| **ScaleGrad w/ tri-gram blocking** |
| --- |
| Two horses are standing in a field. The field is covered with green grass and weeds. The horse is brown and white. The horses are brown and white. There is a large tree behind the horses. The sky is blue and clear. There are many trees in the background. The grass is green. The sun is shining brightly. The sky is mostly cloudy. |

Table 21: Example 2 of generated paragraphs for different training objectives on Visual Genome testset.

| **Reference** |
| --- |
| There is a white and yellow train with red doors next to a red and yellow train. They are on top of the railroad tracks. There is a brick walkway next to the trains. |
| **MLE w/o tri-gram blocking** |
| A train is on the tracks. The train is yellow and yellow. The train is red and white . The train is red and white. The train is red and white . The train is on the train. The train is yellow and yellow. The train is blue and white . The train is on the tracks. The train is yellow and yellow . The train is blue and white. The train is blue and white. The train is on the tracks . The train is yellow. The train is blue. The train is blue. The train is blue. The train is blue. The train is blue. The train is blue. The train is blue. The train is blue. The train is blue. The train is blue. |
| **MLE w/ tri-gram blocking** |
| A train is on the tracks. The train is yellow and yellow. The train has a yellow stripe on the front. The front of the train is red. The sky is blue. There are many windows on the train. There is a train on the platform. The platform is made of metal. The is a platform on the side of the train. The are many tracks on the train. There are wires on the ground. There is a building behind the train tracks. There is a large building behind the train. |
| **ScaleGrad w/ tri-gram blocking** |
| A train is on the tracks. There are two sets of tracks next to the train. The train is white and yellow. There is a large white building behind the trains. |

Table 22: Example 1 of summaries generated by the model trained with different training objective on CNN/DM testset.

| **Reference** |
| --- |
| report by dick marty, lead investigator for council of europe, gives bleak description of secret prisons run by cia in poland and romania from 2003 to 2006. marty says he relied heavily on information from cia agents. critics in poland and romania attack marty's use of anonymous sources. |
| **MLE** |
| swiss sen dick marty reports that central intelligence agency operates secret prisons run by american in poland and romania from 2003 to 2006. says prison were operated exclusively by americans in poland and. |
| **Unlikelihood** |
| swiss sen dick marty reports that secret prisons run by central intelligence agency in eastern europe, with information he says is gleaned from anonymous intelligence agents. report is prepared by swiss senator investigating cia operations for council of europe, 46 - nation rights group. scathing report says prison were operated exclusively by americans in poland and romania from 2003 to 2006. |
| **ScaleGrad** |
| dick marty, swiss senator investigating cia operations for council of europe, gives bleak description of secret prisons run by central intelligence agency in eastern europe, with information he says is gleaned from anonymous intelligence agents. report says prisons were operated exclusively by americans in poland and romania from 2003 to 2006. |

Table 23: Example 2 of summaries generated by the model trained with different training objective on CNN/DM testset.

| Reference |
| --- |
| middleton wrote about eating whale for a national newspaper travel story. she described eating it at the #123-a-night juvet landscape hotel. whale and dolphin conservation group said the news was 'disappointing'. the wdc said ms.middleton should have gone whale watching instead. |
| **MLE** |
| the 31-year-old has been slammed for 'promoting the cruel and unnecessary whaling industry'. the 31-year-old boasted of eating minke whale in norway, to the horror of environmental campaigners who say she is helping promote an 'unimaginably cruel' trade. the hunting and importation of whale meat is banned in britain , as it is in most parts of the world. |
| **Unlikelihood** |
| the 31-year-old boasted of eating minke whale at juvet landscape hotel in norway. the 31-year-old has been slammed for 'promoting the cruel and unnecessary whaling industry '¡q¿environmental campaigners say she is helping promote an ' unimaginably cruel ' trade. |
| **ScaleGrad** |
| duchess of cambridge's brother-in-law has led a personal crusade against poaching and protecting wildlife. pippa middleton boasted of eating minke whale in norway, conservation group said she is helping promote 'promoting the cruel and unnecessary whaling industry'. |

Table 24: Example 1 of summaries generated by the model trained with different training objective on NYT50 testset.

| Reference |
| --- |
| protesters angry over bribery scandal involving state-run oil company petrobras. brazilian president dilma rousseff also is struggling with an economic downturn. |
| **MLE** |
| protesters are calling for president dilma rousseff to be impeached. rousseff's supporters call for the president to be impeachment¡q¿they say there hasn't been any evidence she was involved in the corruption scandal. |
| **Unlikelihood** |
| protesters are calling for president dilma rousseff to be impeached. there is a number of issues at play. one of the biggest is an investigation into a multimillion-dollar kickback scheme at the state-run oil company petrobras. |
| **ScaleGrad** |
| president dilma rousseff is facing an economic downturn and a massive bribery scandal. rousseff has defended her right to protest and acknowledged the need to clean up corruption at petrobras. |

Table 25: Example 2 of summaries generated by the model trained with different training objective on NYT50 testset.

| Reference |
| --- |
| wollemi pine, which flourished 200 million years ago, is available to gardeners and can be grown indoors or outdoors. thought to extinct, it was discovered in 1994 in wollemi national park in australia. botanists took cuttings from wild colony of trees and began propagating baby pines and sending them to botanic gardens around world. |
| **MLE** |
| new york city plant - o - ramachan pine, which grew among dinosaur 200 million years ago, is named after dr david noble, who rappelled into deep gorge in blue mountains of wollemi national park, ten miles west of sydney, australia. photos. map. pine was discovered until 1994, when bush delay and national parks officer david noble tried rappelled into woods of wollemi, which he had never seen before. scientists had evidence of wollemi pine's existence, but only from fossils. |
| **Unlikelihood** |
| nicholas d kristof op-ed column on plight of wollemi pine, one of oldest on earth who grew among dinosaur 200 million years ago. notes that it was not discovered until 1994, when bush rescue officer david noble rappelled into deep gorge in blue mountains of sydney, australia, and came back up with branch of tall tree that he never saw before. |
| **ScaleGrad** |
| nicholas d kristof op-ed column on new species of wollemi pine, which grew among dinosaur 200 million years ago and was discovered until 1994, when bush delay and national parks officer david noble rappelled into deep gorge in blue mountains of wollemi national park, 120 miles west of sydney, australia, and discovered new species, wollemia nobilis, named after noble. |

Table 26: Example of generated continuations for different training objectives on PTB testset.

| Prefix |
| --- |
| last resort who were criticized after the 1987 crash – once again couldn't handle the selling pressure. Big investment banks refused to step up to the plate to support the beleaguered floor traders by buying big blocks of stock, traders say. Heavy selling |
| **MLE** |
| pressure was also felt by the Wall Street banks. The Wall Street banks were forced to sell more stock to hedge funds and other investors, and to buy more stock to hedge against losses. The Wall Street banks were forced to sell more stock to hedge funds and other investors, and to buy more stock to hedge against losses. The Wall Street banks were forced to sell more stock to hedge funds and other investors, and to buy more stock to hedge against losses. |
| **Unlikelihood** |
| pressure was also felt by the banks themselves. In 1987, the New York Federal Reserve Bank, which had been the largest bank in the United States since the 1930s, was forced to sell $1.5 billion worth of stock. The bank's stock price fell to $0.001, and it was forced to close its New York branch. The bank's stock price fell to $0.001, and it was forced to close its New |
| **ScaleGrad** |
| by the big banks led to a sharp drop in prices, and the market collapsed. In 1992, the Federal Reserve Bank of New York issued a report on the financial crisis, which concluded that the Fed's actions had not been sufficient to prevent the collapse of the housing market. The report recommended that the Fed increase its balance sheet by $1.5 trillion (equivalent to about $2.5 trillion today), but this was not implemented. |

Table 27: Example of generated continuations for different training objectives on movie reviews from IMDB dataset.

| **Prefix** |
| --- |
| One of the other reviewers has mentioned that after watching just 1 Oz episode you'll be hooked. They are right, as this is exactly what happened with me. The first thing that struck me about Oz was its brutality and unflinching scenes of violence. |
| **MLE** |
| I was not expecting anything like that in a show about a family. I was expecting a show about a family, but I was expecting something more. I was expecting a show about a family, but I was expecting something more. I was expecting a show about a family, but I was expecting something more. I was expecting a show about a family, but I was expecting something more. I was expecting a show about a family, but I was expecting something more. I was expecting a show |
| **Unlikelihood** |
| I was immediately hooked. The violence was so graphic that I was completely taken aback by how it was handled. I was also struck by the fact that the show was so well-written that it was almost impossible to not be entertained. The second thing that struck me about Oz was its ability to make me laugh. I was so taken aback by the violence that I couldn 't help but laugh. The violence was so graphic that I was completely taken aback by how it |
| **ScaleGrad** |
| It was also very funny, but not funny enough to make me laugh out loud. The second thing that struck me was the fact that the show's characters were so well developed that they were able to carry their own weight without being annoying or annoying. In its original broadcast, "Oz" was watched by 9.5 million viewers, according to Nielsen ratings. It was the highest rated |

