# OpenReview forum: "Straight to the Gradient: Learning to Use Novel Tokens for Neural Text Generation"
_ICLR.cc/2021/Conference — Reject_

### Official Review · AnonReviewer1 · 2020-10-27
**Effective softmax renormalization is reducing degeneracy in neural text generation with MLE training**

**Rating:** 6
**Confidence:** 4

**Review:**

## Summary

This work proposes an effective modification of language model token-level distribution during the training which prevents some forms of degeneration such as repetitions and dullness. The approach is based on the idea of encouraging the model to use tokens which were not observed in the previous context so far. In other words, this method changes softmax distribution such that unseen/novel tokens is being rescaled with a given hyper-parameter $\gamma$ (eq.4). Authors conduct several experiments using different tasks such as open-ended generation, image captioning and abstractive text summarization. As a result they confirm substantial improvement over the standard mle training and **token-level** unlikelihood training. In addition to analysis of their method, authors discuss a potential issue of unlikelihood training criterion and how their approach avoids this issue.

## Strong points

1. Main method is easy to understand and I believe is easy to implement: this work can be a motivating example for future research towards degenerative text generation. From my understanding there are some interesting future work such as setting individial $\gamma$ for some tokens, specific masks for novel token sets etc.

2. Large-scale experiments **with some human evaluation**. I enjoyed seeing improvements on multiple tasks including summarization (with automatic metrics at least). In addition, analysis of stochastic decoding used in directed generation is meaningful and highlights the importance of this work. Appendix includes detailed description of each experiment protocol including the protocol of human evaluation. Convincing examples of generated continuations are given in the appendix.

3. Code with implemented method and experiments is provided: code is based on fairseq custom module which makes it relatively easy to extend and do more research with it.

## Weak points

1. Misleading comparison choice: authors claim to compare their approach with unlikelihood training (UL) and choose **token-level** UL loss without even mentioning the existence of **sequence-level** UL loss which works better based on the original paper. In fact, the whole narrative looks like sequence-level version does not exist. Simply stating that ScaleGrad is being compared with token-level UL (which works worse than sequence-level) would make future conclusions more clear.

2. Some relevant work got completely ignored. I am not aware of the full variety of prior work for this popular problem these days, but there is one i am aware of: https://arxiv.org/pdf/2003.11963.pdf, where authors *do similar gradient analysis* as here. If this one is missed, I wonder what else may be missing in the related work here.

3. No human evaluation for text summarization. Given known weakness of automatic metrics in text summarization task and the fact that authors did human eval for text completion, I wounder why they decided to exclude it from here (I can totally see budget limitation as one of the factors, and saying this explicitly would be helpful).

4. The potential issue of UL (sec. 5.4) does not look convincing. From my understanding the main line there is "*UL essentially rejects the ground truth token in such special cases (subject to the choice of the value of $\alpha$).*". This statement on its own is not clear to me and seems to be disconnected from the previous one: "*In this case, the norm increases as pk increases, which contradicts with the optimization principle.*" I agree that in this case ($\alpha=1, p_\text{neg}>0.5$) the norm increases as $p_k$ (prob of ground truth token) increases, but I don't see any problem or contradiction here. From my understanding when $p_\text{neg}$ goes above some threshold, then norm of the gradient of $p_k$ is growing proportionally to $p_k$. Keep in mind that as $p_k$ is increasing, $p_\text{neg} > \frac{1}{1+\alpha}$ is eventually dissatisfied (because of softmax property), i.e. I don't see any issue. Would be great if authors can elaborate more about this.

## Recommendation

Overall I vote for accepting this work as long as main concerns will be addressed/discussed. This is a decent approach with a strong experimental evidence and it will be useful for text generation community in the future research. I would be even more satisfied if authors can discuss / clarify / address the points I highlighted above.

## Comments and questions

1. stochastic beam search was mentioned as one of the efforts to solve text generation issue, but I believe it is more about doing sampling without replacement on the sequence-level. I am curious if authors may provide some perspective on how stochastic beam may reduce the degeneracy (e.g. compared to simple beam search).

2. In sec. 2.1 teacher forcing is described as being used "*used to train neural text generation models for faster convergence.*". I wonder how can one use MLE criterion on the token level without teacher forcing? E.g. if we use predicted token as the context in the next time step, then we have no target truth token for the next time step (since the context is changed). In other words I think that teacher forcing is essential if we aim to maximize the probability of training sequences.

3. In section 2.2: "*thus reformalizing the probability distribution*", this wording reformalizing sounds a bit weird to me, but it is clear what authors had in mind.

4. Did you think about combining UL loss (both seq level and token level) with ScaleGrad? From my understanding it is possible since ScaleGrad emphasize novel tokens, and this softmax from scalegrad may be used in the UL loss, which may help even further! Importantly, sequence-level UL would allow to use ScaleGrad on the sequence-level, since there is no need for ground truth target there, and while penalizing the repeating words with UL, ScaleGrad would emphasize the novel words (sounds promising to me). Overall the paper narrative looks as combating the UL method (with some misleading gaps about token vs. sequence level), but to me it looks like they may work together!

---

### Official Review · AnonReviewer3 · 2020-10-28
**Review: Straight to the Gradient: Learning to Use Novel Tokens for Neural Text Generation**

**Rating:** 5
**Confidence:** 5

**Review:**

The authors propose to modify a language model's token-level distributions by rescaling the output probability of tokens that do no appear in the context ('novel tokens'). The authors show improvements over MLE and token-level unlikelihood in terms of repetition, with increases in perplexity.


#### Clarity and significance
- **ScaleGrad motivation**. There are many different ways to change the gradient, e.g. any regularization function, any scaling of the output probabilities, or even gradient clipping modifies the gradients that are used to update the model. As a result, the presentation of their method as "a modification straight to the gradient of the loss function" seems odd, and the name ScaleGrad suggests that they are proposing the general notion of rescaling gradients. Instead, they propose to scale a novel set's output probabilities then renormalize.

- **Specific solution**. The method is specifically designed around the 'novel set', which could limit its significance. The authors speculate that they can alter the novel set (e.g. for importance or factual correctness), but this appears to be nontrivial.

- **Unlikelihood discussion**. The discussion in 5.4 deals with the case of $p_{neg}>0.5$, meaning that the probability of the ground-truth token $p_*$ is $<0.5$ (due to normalization). If I'm understanding their argument, the authors argue that the resulting gradient contradicts the fact that the gradient should go to zero at an optimum. However, *the model is not at an optimum* if $p_<0.5$. Could the authors clarify the statements "the model is not to learn to predict the ground truth tokens correctly", "contradicts the optimization principle", and "essentially rejects the ground truth token"?

- **Method for promoting novelty**. It's unclear why this specific method (renormalizing over the novel set) is the best or simplest method for promoting novelty. A downside is that we no longer know which objective the model is optimizing. In the appendix, the authors discuss a variant that uses an additional loss (Section I), yet do not perform an empirical comparison with that or other 'novelty promoting' variations. They argue in Section I that this suffers 'the similar issue as the UL loss', but that issue was unclear (see point above).

Overall I'm borderline on this paper: the authors do perform a lot of experiments and show improvements, but I'm hesitant that scaling novel tokens and renormalizing the model's output distribution is significant.

---

### Official Review · AnonReviewer4 · 2020-10-29
**Comparisons are Incomplete and Sometimes Occluded**

**Rating:** 4
**Confidence:** 4

**Review:**

**I have updated this review after noting the authors’ detailed response.**

This paper focuses on the problem of “Neural Text Degeneration”—where text sampled from a language model can either be too repetitive and bland or too random and nonsensical. The authors focus largely on the former problem, proposing a finetuning loss that specifically incentivizes the use of tokens that have not yet been decoded in the given document. The authors test whether this improves repetition and unique token coverage with greedy decoding in open-ended generation. A small human study is conducted and the proposed method, ScaleGrad, is found to outperform MLE and Unlikelihood Training (UT). Similarly good results are obtained on Image Captioning with and without trigram repetition blocking. On Abstractive Summarization BeamSearch is used and again outperform MLE and UT. Analysis attempts to make comparisons across different decoding strategies, though coverage of different variations is limited. The authors argue that stochastic decoding is outperformed by ScaleGrad, though they note that trigram blocking still helps ScaleGrad. Multiple hyperparameter settings are shown, with some analysis on how gamma can be chosen to get a desired behavior. Finally, the authors analyze why UT may not be as effective: it penalizes gold repetitions too much and does little for other tokens.

Strength:
- The results are good for greedy decoding
- The method is well motivated and well explained
- The analysis regarding Unlikelihood Training is interesting

Weaknesses
-  The results shown do not make proper comparisons across models, baselines, and hyperaparameters over all tasks.
- Results for stochastic decoding should have been shown across tasks.
- Despite citing the need for awkward rules such as trigram repetition blocking as a reason to propose ScaleGrad, trigram repetition blocking still helps significantly.
- Some details are hidden away in the appendix, which I had to read thoroughly in order to fully understand the comparison.

I recommend to reject this paper, because the experimental comparisons are not quite fair and because of implicit argumentation about what Greedy decoding can or should do that is never made explicit.

The following two paragraphs are obsolete, because the authors shared experimental results from a larger set of experiments.
> The results in Table 1—which show the main metrics of interest on open-ended generation—are missing two key points of comparison: ScaleGrad is only show with gamma=0.2, even though gamma=0.5 & gamma=0.8 are used for the rest of the experiments, giving us little idea of how these metrics change over hyperparameter settings. This is despite the fact that two hyperparameter options for Unlikelihood Training are shown. In a footnote on page 6, for directed generation, the authors state “Although UL was originally proposed for open-ended generation, it is applicable to directed generation. We did  the same scale hyper-parameter search for UL. Details can be seen in Appendix E.” However, in Appendix E two hyperparameter settings for alpha are shown, the same two as used in Table 1, but two hyperparameter settings for gamma in ScaleGrad are shown _neither of which are shown in Table 1_ nor are repetition or uniqueness numbers shown for these hyperparameters settings anywhere in the paper or the appendices. This makes me question whether the improvements shown in Table 1 hold across hyperparameter settings as the authors claim in their analysis of Figure 1.

> However, Figure 1 is missing necessary data points and comparison. First of all gamma=0.2 is not shown, though at least gamma=0.1,0.3 are so it can be somewhat inferred. That is suboptimal, but this graph does not even go up to gamma=0.8, which is what is used in the Abstractive Text Summarization experiment! Furthermore, the number in Figure 1 (b) cannot be directly compared to other decoding methods, because they are an average of repetition metrics shown in Table 1. Luckily, Figure 1 (c) can be compared, and if cross-referenced with Table 1, shows that Unlikelihood Training does better than ScaleGrad with a higher gamma. However, Figure 1 has no data on either Unlikelihood Training or a human baseline. It really should not be necessary to go looking through Table 1, Figure 1, and Appendix F to see that Unlikelihood Training is outperforming ScaleGrad on some metrics. Worse, the data presented in Figure 1 (b) actually makes comparison impossible, which makes me uncomfortable about the universally positive results in Table 1.

On page 4 the authors write “Following Welleck et al. (2020), we apply greedy decoding in our experiments in this section. This allows us to evaluate the modeling capability exclusively.” We will get into the matter of comparison to Welleck et al. 2020, but I would like to begin by addressing whether Greedy Decoding is a neutral choice that only tests modeling capability, because it is clearly not. There is a spectrum of generation algorithms between probability maximization and straight-forward sampling. Greedy is closer to probability maximization, but it only maximizes local probabilities (Meister et al., 2020) and inevitably comes-up with lower probability outputs than Beam Search or Bound & Branch (Stahlberg & Byrne, 2019). Welleck et al., 2020 show that Greedy Decoding results in better text along their proposed metrics for open-ended generation.

Since Greedy Decoding is not a “neutral” choice, I do not believe it is appropriate to exclude stochastic decoding baselines from the given comparisons. Stochastic decoding algorithms such as sampling, top-k sampling, and Nucleus Sampling usually do very well on repetition and uniqueness metrics. Indeed, they can be seen to outperform all the other models on Table 16 in Appendix H.

In the analysis section, tables are quite limited in their coverage. In Table 6 no comparisons are made to systems that have not been trained with ScaleGrad, and these algorithms were not reported on in Table 1 so no comparison can properly be made even if the reader goes searching for the data.  In Table 8, Unlikelihood Training is not included in the comparison even though it does very similarly to ScaleGrad on the same task in Table 5. Finally, Table 5 shows that trigram blocking still helps significantly on ScaleGrad trained systems. This is understandable, but disappointing since getting rid of these kind of rules is described as the reason for proposing ScaleGrad.

Altogether, I feel the comparisons made in this paper are not quite convincing and the argument about why Greedy decoding, a deterministic algorithm, should even be able to match the properties of a large, noisy distribution is not properly fleshed-out.

Meister, Clara, Ryan Cotterell, and Tim Vieira. "If Beam Search Is the Answer, What Was the Question?." Proceedings of the 2020 Conference on Empirical Methods in Natural Language Processing (EMNLP). 2020.

Stahlberg, Felix, and Bill Byrne. "On NMT Search Errors and Model Errors: Cat Got Your Tongue?." Proceedings of the 2019 Conference on Empirical Methods in Natural Language Processing and the 9th International Joint Conference on Natural Language Processing (EMNLP-IJCNLP). 2019.

---

### Official Review · AnonReviewer2 · 2020-10-29
**New technique for encouraging novel tokens in text generation.**

**Rating:** 6
**Confidence:** 3

**Review:**

The paper presents a  technique to encourage generating certain tokens (i.e. non-repetitive ones) in text generation. The idea is to scale the softmax probability  for certain words (in the novel set) by a factor of gamma. The authors show how this affects learning by deriving the effect on the gradient.

Many experiments are presented both on open ended generation (language modeling, story telling) as well as abstractive text summarization to justify the method. The model seems to encourage more diversity than unlikelihood training in open ended generation (while still maintaining a lower perplexity). The gains in summarization are marginal.

I have two questions:
(1) How are lambda and alpha connected to \gamma in Section 3? This make the method section clearer.
(2) How much is the model discouraged from generating stop words like "the" or "a" (and how does this affect fluency)

Pros:
-Well justified and simple method to solve a relevant problem in text generation.
-Lots of experiments, gains in open ended generation seem decent.

Cons:
-Gains on summarization are marginal / non-existent suggesting that this is not as large of a problem for more constrained tasks.
-Some clarity on the questions above would be helpful.

---

### Decision · Program_Chairs · 2021-01-07
**Final Decision**

**Decision:**

Reject

**Comment:**

This paper proposes ScaleGrad, a simple technique to encourage generating non-repetitive tokens for text generation tasks. The key idea is to modify a language model's token-level distributions by rescaling the softmax probability for certain words (in the novel set) by a factor of $\gamma$. Experiments show that ScaleGrad outperforms MLE and Unlikelihood Training (UT).

This paper receives 2 reject and 2 accept recommendations. Most of the reviewers have provided very detailed comments, and the authors have also provided very long and detailed responses. On one hand, all the reviewers agree that the experiments are comprehensive, and the motivation of the proposed method is clear.

On the other hand, several concerns still exist after the rebuttal, namely, hand-wavy arguments and inconsistent experimental protocol. (i) The empirical evidence in the experiments is not convincing enough. It makes reviewers more reluctant about the approach after seeing more experimental results during the discussion. That is, for different tasks, different $\gamma$s are used, while the hyper-parameters used for other methods seem to be default values. This makes reviewers hesitant that scaling novel tokens and renormalizing the model's output distribution is really significant. (ii) Another minor concern is that the discussion on the potential issue of UL (sec. 5.4) does not look convincing. (iii) After reading the paper, the AC also feels that the novelty of the proposed method might be a little bit limited.

The rebuttal unfortunately did not fully address the reviewers' main concerns. On balance, the AC regrets that the paper cannot be recommended for acceptance at this time. The authors are encouraged to consider the reviewers' comments when revising the paper for submission elsewhere.